# H_2_S in Critical Illness—A New Horizon for Sodium Thiosulfate?

**DOI:** 10.3390/biom12040543

**Published:** 2022-04-04

**Authors:** Tamara Merz, Oscar McCook, Cosima Brucker, Christiane Waller, Enrico Calzia, Peter Radermacher, Thomas Datzmann

**Affiliations:** 1Institute for Anesthesiological Pathophysiology and Process Engineering, Ulm University Medical Center, 89081 Ulm, Germany; oscar.mccook@uni-ulm.de (O.M.); enrico.calzia@uni-ulm.de (E.C.); peter.radermacher@uni-ulm.de (P.R.); 2Clinic for Anesthesiology and Intensive Care, Ulm University Medical Center, 89081 Ulm, Germany; thomas.datzmann@uni-ulm.de; 3Department of Gynecology and Obstetrics, Nuremberg General Hospital, Paracelsus Medical University, 90419 Nuremberg, Germany; cosima.brucker@klinikum-nuernberg.de; 4Department of Psychosomatic Medicine and Psychotherapy, Nuremberg General Hospital, Paracelsus Medical University, 90419 Nuremberg, Germany; christiane.waller@klinikum-nuernberg.de

**Keywords:** ischemia/reperfusion injury, inflammation, oxidative stress

## Abstract

Ever since the discovery of endogenous H_2_S and the identification of its cytoprotective properties, efforts have been made to develop strategies to use H_2_S as a therapeutic agent. The ability of H_2_S to regulate vascular tone, inflammation, oxidative stress, and apoptosis might be particularly useful in the therapeutic management of critical illness. However, neither the inhalation of gaseous H_2_S, nor the administration of inorganic H_2_S-releasing salts or slow-releasing H_2_S-donors are feasible for clinical use. Na_2_S_2_O_3_ is a clinically approved compound with a good safety profile and is able to release H_2_S, in particular under hypoxic conditions. Pre-clinical studies show promise for Na_2_S_2_O_3_ in the acute management of critical illness. A current clinical trial is investigating the therapeutic potential for Na_2_S_2_O_3_ in myocardial infarct. Pre-eclampsia and COVID-19 pneumonia might be relevant targets for future clinical trials.

## 1. Introduction

This review summarizes the current evidence for the therapeutic potential of sodium thiosulfate (Na_2_S_2_O_3_), a clinically approved H_2_S donor with minimal side effects, in critical illness. H_2_S has a variety of biological roles, such as the regulation of vascular tone, as well as anti-oxidant and anti-inflammatory properties, which could exert a potential therapeutic benefit in intensive care. However, to date, the narrow therapeutic window and potential for toxic adverse effects have prevented the application of gaseous inhaled H_2_S in the clinical setting [1,2,3,4,5]. Other strategies for H_2_S administration, such as H_2_S-releasing salts (NaHS and Na_2_S) and specifically developed H_2_S slow-releasing donors (such as GYY4137 and AP39) with a better pharmacokinetic profile, have similar limitations for clinical use [5,6,7,8]. Na_2_S_2_O_3_ is a detoxifying agent, which is clinically approved as an antidote for cyanide poisoning as well as chronic renal failure-induced calciphylaxis [9] and cisplatin overdose [10]. Na_2_S_2_O_3_ can induce metabolic acidosis, but otherwise no major side effects have been reported [11,12,13], thus Na_2_S_2_O_3_ is a promising H_2_S donor for clinical use. The thiosulfate anion (S_2_O_3_^2−^) is an endogenous oxidation product of H_2_S degradation, and, in turn, can serve as a source of H_2_S [14]. In particular, under hypoxic conditions, 3-mercaptopyruvate sulfurtansferase (3-MST) and rhodanese have been reported to be able to facilitate sulfide release from thiosulfate [14].

## 2. Biology of H_2_S and S_2_O_3_^2−^

Originally, H_2_S was known for its toxicity to humans, due to its reversible inhibition of cytochrome-c-oxidase, Complex IV of the mitochondrial electron transfer system (ETS), blocking its O_2_ binding site. About 30 years ago, endogenous H_2_S was first detected [15] and described as a gaseous neurotransmitter [16], and since then three different enzymes have been identified to be able to release H_2_S endogenously: cystathionine-γ-lyase (CSE), cystathionine-β-synthase (CBS), and 3-MST. 3-MST produces H_2_S from 3-mercaptopyruvate, which is generated from cysteine (see Figure 1). The sulfur atom is transported from 3-mercaptopyruvate by MST in the form of sulfane sulfur (MST-SSH). It was shown that H_2_S could be released in the presence of dithiols (dihydrolipoic acid (DHLA), DTT, thioredoxin (Trx)); however, in the presence of monothiols (GSH, cysteine), no release of H_2_S was observed [17]. H_2_S oxidation represents another level of the regulation of endogenous H_2_S availability, besides its endogenous production. In the mitochondria, H_2_S can be oxidized by sulfide quinone oxidoreductase (SQR), which can contribute to the pool of electrons processed in the ETS [14,18,19]. Thus, small amounts of H_2_S (nanomolar up to low micromolar concentrations [20]) can actually stimulate mitochondrial respiration. A physiological acceptor of sulfane sulfur from SQR-SSH has not been identified unequivocally, yet. Some authors have postulated that human SQR utilizes sulfite as persulfide acceptor, yielding thiosulfate as a product. Other authors have demonstrated that in addition to sulfite, GSH functions as a persulfide acceptor for human SQR, leading to GSSH [21]. It is also not excluded that there are SQR persulfide acceptors other than sulfite and GSH (e.g., DHLA, Trx, cysteine), which, after accepting sulfane sulfur, can be reduced by GSH [22]. Hydropersulfides (-SSH) are further oxidized by sulfur dioxygenase and sulfur transferase to thiosulfate [14]. Hemoproteins might also contribute to thiosulfate production from H_2_S [23]. Thiosulfate is further oxidized into sulfate. In turn, in particular under hypoxic and reducing conditions, H_2_S can be regenerated from exogenously administered thiosulfate by rhondanese and, reportedly, 3-MST [14] (see Figure 1), which is not surprising when considering that both thiosulfate and 3-mercaptopyruvate are, chemically speaking, sulfane sulfurs.

Sulfide oxidation seems to be a critical process in cellular oxygen sensing [24]. Under normoxic conditions, sulfide is oxidized continuously into thiosulfate in the mitochondria [25], whereas under hypoxic conditions, sulfide oxidation is limited and cellular H_2_S concentrations rise [19,25]. Hypoxic conditions additionally favor endogenous enzymatic H_2_S production, which in turn seems to mediate intracellular signaling in response to hypoxia, such as hypoxic vasoconstriction and catecholamine release [25]. Furthermore, as mentioned above, thiosulfate itself can serve as a source of H_2_S under hypoxic conditions, but not during normoxia [14].

We investigated the effects of Na_2_S_2_O_3_ on mitochondrial respiration in an in vitro experiment in cultured rat primary cortical neurons. Cells were harvested after 4 h of incubation with either 4, 20, or 100 mM Na_2_S_2_O_3_ for the assessment of mitochondrial oxygen consumption (JO_2_) at the maximum respiratory activity of the electron transfer system in the uncoupled state. Indeed, we were able to identify a differential dose-dependent effect of Na_2_S_2_O_3_ on mitochondrial respiration. It is likely that the cells were able to release H_2_S from the administered Na_2_S_2_O_3_, which (at a dose of 4 mM Na_2_S_2_O_3_) either stimulated mitochondrial respiration as an electron donor via Complex II or (at a dose of 100 mM Na_2_S_2_O_3_) inhibited Complex IV (see Figure 2).

Given the tight interplay of O_2_ and H_2_S availability, it is not surprising that H_2_S plays a role in mediating many consequences of hypoxia: e.g., regulation of vascular tone, inflammation, oxidative stress, and apoptosis, all of which are affected by critical illness. For example, hemorrhagic shock-induced tissue ischemia and hypoxemia can trigger systemic hyper-inflammation [26], and reperfusion can cause ischemia/reperfusion injury (I/R), which contributes to systemic inflammation, oxidative stress, and multiple organ failure [27]. Even though the potential vaso-dilatory effects of H_2_S donors could aggravate shock-induced hypotension [7], other beneficial H_2_S effects might outweigh that risk. Still, the optimal timing and dosing window of H_2_S donors in these conditions must be evaluated carefully. Exogenous H_2_S administration can be facilitated by using inorganic H_2_S-releasing salts (NaSH and Na_2_S), slow-releasing H_2_S donors (GYY4137, AP39), and pre-existing clinically approved compounds, which have been recently identified to be able to release H_2_S (ammonium tetrathiomolybdate (ATTM), Na_2_S_2_O_3_). H_2_S-releasing salts cause rapidly increasing, potentially toxic, peak H_2_S concentrations, which dissipate quickly [28], can have pro-inflammatory effects [29], and damage the mitochondria [20,30]. Even when these peak concentrations are prevented, the therapeutic window for H_2_S-releasing salts is very narrow, which is why they are unsuitable for clinical use [4,5,6,31]. In contrast, slow H_2_S-releasing donors seem to have different effects, in that they rather seem to ameliorate inflammation [29]. However, translationally relevant in vivo results for slow-releasing H_2_S donors also render them unlikely for clinical development, since they have failed to exert organ protection and/or even have adverse effects in these models so far [7,8], in spite of many promising pre-clinical studies in non-resuscitated rodents. ATTM showed organ protection in in vivo models of I/R [32,33]. However, it is not clear if ATTM is superior to standard treatment, because these studies did not include standard ICU measures. Na_2_S_2_O_3_ is a clinically available compound with a good safety profile [12] and is reportedly able to release H_2_S, in particular, under hypoxic conditions [14]. Further biochemical effects of STS are summarized in Figure 3. Its potential to reduce oxidized glutathione and its effects on vascular tone might be of particular importance for therapeutic approaches in critical care. Thus, Na_2_S_2_O_3_ is a promising candidate to bring H_2_S therapy in critical illness to the clinic. The following section will explore available literature reports of exogenous Na_2_S_2_O_3_ administration in experimental models of critical illness.

## 3. Experimental Evidence for Beneficial Effects of Na_2_S_2_O_3_ in Animal Models of Critical Illness

Animal studies investigating the effects of Na_2_S_2_O_3_ in models of critical illness are summarized in Table 1.

### 3.1. Langendorff Heart Model and In Vivo I/R Experiments

In seven different ex vivo studies using the Langendorff rat heart model (i.e., isolated heart with retrograde perfusion with a nutrient-rich oxygenated buffer), with 30 min ischemia followed by 60 min reperfusion, a variety of beneficial effects of different modes of Na_2_S_2_O_3_ administration have been reported. Both pre- [39,40,41,42] and post-conditioning [43] or Na_2_S_2_O_3_ administration starting with reperfusion [41,42,44] were reported to reduce cardiac injury, apoptosis [39,43], inflammatory markers [39], and oxidative stress, [39,42,43,44] and to protect cardiac mitochondria [39,40,42,43,44], resulting in improved cardiac contractility [40,44] in comparison to non-Na_2_S_2_O_3_-treated hearts. The same group of scientists was also able to confirm the beneficial pre-conditioning effects of Na_2_S_2_O_3_ in in vivo rat model of myocardial I/R, i.e., left anterior descending artery ligation [39] and isoproterenol-induced myocardial infarction (MI) [45]. The beneficial effects of Na_2_S_2_O_3_ in MI were also demonstrated in dogs [46]. In one of the in vivo rat MI studies, an additional beneficial effect on the brain—reduced apoptosis and oxidative stress associated with improved mitochondrial function—was determined [45]. Marutani et al. also reported a benefit of repeated i.p. Na_2_S_2_O_3_ administration on the murine brain after cerebral I/R by bilateral common carotid artery occlusion: improved 20-day survival and better neuronal function in the treated animals [47].

**Table 1 biomolecules-12-00543-t001:** Summary of *in vivo/ex vivo* studies investigating the effects of Na_2_S_2_O_3_ in models of critical illness.

First Author	Animal Model	Condition	Na_2_S_2_O_3_ Administration	Effects of Na_2_S_2_O_3_ (Compared to Vehicle)
Mouse models				
Tokuda 2012 [48]	Male C57BL/6J mice, aged 8–10 weeks	Endotoxemia: LPS (10 mg/kg i.p.) with 1 mL of saline (i.p.) at 0, 6 and 24 h after LPS	1 or 2 g/kg (i.p.) immediately after LPS	improved survival
Shirozu 2014 [49]	Male C57BL/6J mice, aged 8–12 weeks	Acute liver failure and endotoxemia: D-Galactosamine (300 mg/kg) and 1 mg/kg LPS (i.p.) with 1 mL of saline (i.p.) at 0, 6 and 24 h after LPS	2 g/kg (i.p.) at 30 min before and 3 h after LPS	liver injury ↓, antioxidant and anti-apoptotic effects, preserved mitochondrial membrane potential
Sakaguchi 2014 [50]	Male C57BL6J mice, aged 8–10 weeks	Endotoxemia: intratracheal LPS (2 mg/kg) Polymicrobial septic shock: CLP with fluid resuscitation	2 g/kg (i.p.) at 0 and 12 h after LPS 0.5 g/kg (i.v.) at 10 min after CLP	cell accumulation and MPO in BALF ↓, lung edema ↓, lung IL-6, TNFα, NOS2, MMP9 ↓, lung NFκB signaling ↓
Marutani 2015 [47]	Male C57BL/6J mice, aged 8–9 weeks	Global cerebral I/R: bilateral common carotid artery occlusion, reperfusion after 40 min	10 mg/kg (i.p.) in 1 mL of 5% dextrose-enriched lactated Ringer’s solution at 10 min after reperfusion and daily for 1 week	improved 20-day survival, neuronal function score ↑ at 24 h after reperfusion
Acero 2017 [51]	Female C57BL/6J mice, aged 2 months	Endotoxemia: LPS (3 mg/kg i.p.)	100, 350, 500, and 750 mg/kg (i.p.) immediately after LPS and at 8, 24, and 32 h	brain IL1-β, COX-2, Iba-1, TSPO ↓ (at 500 mg/kg Na_2_S_2_O_3_) neuroinflammation ↓
Renieris 2021 [52]	Male and female mice, WT and CSE^−/−^ (C57BL/6) aged 7–8 weeks	Sepsis: *P. aeruginosa* (i.p.) infection	2 g/kg (s.c.) daily for four days after infection	survival ↑ in WT mice
Gröger 2022a [53]	Male CSE^−/−^ mice (C57BL/6J.129SvEv), aged 20–25 weeks	Polytrauma: blast wave-induced blunt chest trauma + hemorrhagic shock (1 h), retransfusion and intensive care management (6 h)	i.v. bolus (0.45 mg/g) at start of resuscitation	norepinephrine requirements, lactate ↓, Horowitz index and urine output ↑, lung IL-6 and MCP1 ↓, lung GR and NOS2 ↑, kidney IκBα and HO-1 ↑
Rat models				
Ravindran 2017 [43]	Male Wistar rats Langendorff heart model	30 min ischemia with 60 min reperfusion	1 mM (15 min) post-conditioning	apoptosis ↓, anti-oxidant defense ↑, preserved mitochondrial enzyme activity
Ravindran 2017 b [39]	Langendorff heart model	30 min ischemia with 60 min reperfusion	0.1 mM and 1 mM (10 min) pre-conditioning	myocardial injury, inflammatory cell infiltration, and interstitial oedema ↓
Ravindran 2017 b [39]	Rats	Left Anterior Descending Artery Ligation (30 min ischemia with 2h reperfusion)	1 mM (i.v., 15 min prior to ischemia)	oxidative stress ↓, mitochondrial protection
Ravindran 2018 [54]	Male Wistar rats (aged 6 weeks) with adenine-induced vascular calcification, Langendorff heart model	30 min ischemia with 60 min reperfusion	400 mg/kg orally for 28 days	calcification ↓, I/R-induced cardiac injury ↓ in non-calcified hearts, oxidative stress ↓ in non-calcified hearts, no effect in calcified hearts after I/R
Ravindran 2018 b [44]	Langendorff heart model	30 min ischemia with 60 min reperfusion	1 mM at start of reperfusion	contractility ↑, myocardial injury ↓, mitochondrial enzyme activity ↑, oxidative stress ↓
Ravindran 2019 [40]	Male Wistar rats, 200–250 g Langendorff heart model	30 min ischemia with 60 min reperfusion	1 mM 15 min before ischemia	LV contractility ↑, cardiac injury ↓, recovered ATP production, PGC1α and mitochondrial copy number ↑, mitochondrial proteins ↑, better mitochondrial ultrastructure
Kannan 2019 [41]	Male Wistar rats Langendorff heart model	30 min ischemia with 60 min reperfusion	1 mM at 15 min before I/R 1 mM at reperfusion with(out) additional PAG (CSE/CBS inhibitor)	Infarct size ↓, less pronounced with PAG co-treatment
Ravindran 2020 [45]	Male Wistar rats (250–300 g)	Isoproterenol-induced MI	100 mg/kg (i.p.), 1 h before isoproterenol	cardiac injury ↓, ROS and caspase-3 and 9 ↓ (heart and brain), mitochondrial function ↑ (heart and brain)
Boovarahan 2021 [42]	Male Wistar rats (8–12 weeks, 250–300 g) Langendorff heart model	30 min ischemia with reperfusion	1 mM before ischemia or at reperfusion	Preconditioning protective through ↓oxidative stress, mitochondrial protection Postconditioning protective (different mechanism)
Schulz 2021 [55]	Male Wistar rats (320–380 g)	Polymicrobial sepsis: CASP	1 g/kg (i.p.) immediately after sepsis induction and at 24 h	24 h after sepsis induction: colonic and hepatic microcirculation ↑, mitochondria =,
*large animal models*				
Oksman 1982 [46]	Male and female dogs (15–25 kg)	Tourniquet shock experimental MI (descending coronary artery ligation)	500 µg/kg (i.v., 30 min after removal) 500 mg/kg (i.v., 15 min after ligation)	heart function ↑, blood pressure ↑ heart function ↑, minimal effects on blood pressure
Broner 1989 [56]	New Zealand white rabbits (approx. 3 kg)	*E. coli* septicemia with fluid resuscitation	660 mg/kg (i.v. bolus in combination with other antioxidants, then continuous infusion of 190 mg/kg/h)	well tolerated, no benefit
Datzmann 2020 [13]	Male and female Familial Hypercholesterolemia Bretoncelles Meishan pigs (reduced CSE protein levels), median age 24 months	Hemorrhagic shock (3 h), retransfusion and intensive care management	continuous i.v. infusion at start of retransfusion for 24 h (0.1 g/(kg∙h))	PEEP and Horowitz-Index ↑ (48 h after shock), lung GR expression ↑, pH and BE ↓

Abbreviations: lipopolysaccharide (LPS), intravenous (i.v.), intraperitoneal (i.p.), cecal ligation and puncture (CLP), myeloperoxidase (MPO), bronchoalveolar lavage fluid (BALF), interleukin 6 (IL-6), tumor necrosis factor alpha (TNFα), inducible nitric oxide synthase (NOS2), matrix metalloproteinase 9 (MMP9), monocyte chemoattractant protein 1 (MCP1), glucocorticoid receptor (GR), ischemia-reperfusion injury (I/R), nuclear factor ‘kappa-light-chain-enhancer’ of activated B-cells (NFκB), NFκB inhibitor alpha (IκBα), heme-oxygenase 1 (HO-1), positive end-expiratory pressure (PEEP), base excess (BE), left-ventricular (LV), adenosine -tris-phosphate (ATP), peroxisome proliferator-activated receptor gamma coactivator 1-alpha (PGC1α), myocardial infarction (MI), colon ascendens stent peritonitis (CASP), l-propargylglycine (PAG), cyclo-oxygenase 2 (COX2), microglial marker ionized calcium-binding adapter molecule 1 (Iba-1), 18 kDa translocator protein (TSPO), ↑ higher, ↓ reduced.

### 3.2. Chronic Conditions

In rats with 28 days of oral administration of Na_2_S_2_O_3_ and adenine-induced vascular calcification, Na_2_S_2_O_3_ treatment alleviated the calcification [54]. In this study, hearts from Na_2_S_2_O_3_-treated animals without vascular calcification were also characterized by reduced cardiac injury and diminished oxidative stress after the Langendorff I/R experiments [54]. However, this benefit of Na_2_S_2_O_3_ did not extend to calcified hearts after I/R challenge [54]. Mohan et al. reported a beneficial effect of both oral preventative and therapeutic Na_2_S_2_O_3_ on adenine-induced renal failure, which also rendered the mitochondria of treated animals more resistant to in vitro I/R [57]. Angiotensin-II (Ang-II)-induced hypertension, cardiac hypertrophy, and cardiac fibrosis was also alleviated by regular i.p. injections of Na_2_S_2_O_3_ in rats, which were also associated with a reduction of oxidative stress [58]. In rats with l-NNA-administration, an inhibitor of nitric oxide synthases which causes hypertension and chronic heart disease, two weeks of oral administration of Na_2_S_2_O_3_ ameliorated the hypertension, improved cardiac function [35], and improved glomerular filtration [59]. Interestingly, the authors of these studies did not observe any metabolic acidosis associated with the oral administration of Na_2_S_2_O_3_ in their animals, which is an advantage of oral administration over infusion [35]: patients with i.v. administration of Na_2_S_2_O_3_ have been reported to develop metabolic acidosis [11,60]. This begs the question of if oral administration is the preferable route of administration, even though oral Na_2_S_2_O_3_ bioavailability seems much more limited when compared to i.v. infusion [12]. However, the effects of oral Na_2_S_2_O_3_ on an acute injury in combination with the hypertension were not investigated by these groups. Chronic hypertension as a co-morbidity might worsen the patient’s outcome after critical illness, but is, in itself, not a condition that is normally treated in the intensive care unit. These reported “long-term” benefits show promise for Na_2_S_2_O_3_ as a therapeutic agent but are not feasible for the acute management of critical illness [35,59]. However, it is tempting to speculate that “long-term” effects of oral Na_2_S_2_O_3_ might still be associated with a benefit for the patient after an acute injury.

### 3.3. Endotoxemia and Sepsis

In models of endotoxemia, i.p. Na_2_S_2_O_3_ post-treatment in mice led to improved survival [48], attenuated lung inflammation [50], and ameliorated neuroinflammation [51]. In animals with concomitant acute liver failure and LPS administration, Na_2_S_2_O_3_ treatment decreased liver injury associated with anti-oxidant effects and preserved mitochondrial function [49]. In models of polymicrobial sepsis, Na_2_S_2_O_3_ also had beneficial effects on the murine lung [50] and colonic and hepatic microcirculation in rats [55]. In contrast, in rabbits with *E. coli* septicemia and fluid resuscitation, Na_2_S_2_O_3_ in combination with other antioxidants did not have a benefit [56].

### 3.4. Role of Endogenous H_2_S Enzymes

To date, the role of endogenous H_2_S-producing enzymes for the beneficial Na_2_S_2_O_3_-mediated effects is not entirely clear. In a study with mice with arteriovenous fistula (AVF)-induced chronic heart failure, the authors suggest that Na_2_S_2_O_3_ exerted a benefit by attenuating the AVF-induced loss of CSE expression and thus enhancing endogenous H_2_S generation [61]. In the Langendorff heart model, a co-treatment with the CSE and CBS inhibitor l-propargylglycine (PAG) attenuated Na_2_S_2_O_3_ effects [41]. In *P. aeruginosa*-induced sepsis, daily s.c. Na_2_S_2_O_3_ injections for four days improved survival of wildtype mice, however, there was no survival benefit in CSE^−/−^ mice in the same study [52]. These studies hint towards a critical role for CSE in mediating Na_2_S_2_O_3_ effects. In contrast, in resuscitated CSE^−/−^ mice with blunt chest trauma and hemorrhagic shock, Na_2_S_2_O_3_ clearly reduced the vasopressor requirements needed to achieve hemodynamic targets, thereby attenuating lactic acidosis and both lung and kidney dysfunction [53]. However, in CSE^−/−^ mice with diabetes type I as a co-morbidity undergoing the same kind of injury, this benefit of Na_2_S_2_O_3_ was lost [62] (under review). Wildtype animals were not yet investigated in this context. Furthermore, in a translationally relevant porcine model of hemorrhagic shock with underlying atherosclerosis, Na_2_S_2_O_3_ also exerted a benefit [13]. Here, it is important to point out that atherosclerosis is associated with a down-regulation of the endogenous H_2_S enzymes, which worsens the outcome of these animals from critical injury [63,64,65]. Even though these animals can be considered a large-animal analog to CSE^−/−^ mice, with respect to their lack of CSE, they have reduced CSE expression and not a global knock out as the CSE^−/−^, thus the atherosclerotic pigs might still have residual CSE activity. Still, in the atherosclerotic pigs with hemorrhagic shock and subsequent intensive care management, i.v. administration of Na_2_S_2_O_3_ during the first 24 h of resuscitation was associated with improved lung function at 48 and 72 h after hemorrhagic shock, which was associated with increased levels of glucocorticoid receptor expression in the lung [13]. Strikingly, the clinical benefit of Na_2_S_2_O_3_ on lung function could not be observed in resuscitated cardiovascular healthy pigs after hemorrhagic shock [66] (unpublished data), which, in contrast to some of the results from mouse experiments, suggests a particular benefit from Na_2_S_2_O_3_ for subjects with impaired CSE/impaired endogenous H_2_S availability.

### 3.5. Role of the Interplay of H_2_S and Thiosulfate

Interestingly, in the atherosclerotic pigs, the beneficial effects manifested 24 h after the Na_2_S_2_O_3_ had already been stopped, even though elevated sulfide levels were only detected immediately at the end of Na_2_S_2_O_3_ administration, but not 24 h after the administration [13]. This begs the question of if the elevation of sulfide levels or the thiosulfate itself actually exerts the therapeutic benefit. In one of the abovementioned studies, the authors added Na_2_S_2_O_3_-treated groups only after observing that H_2_S inhalation in their study elevated plasma sulfide as well as thiosulfate levels [48]. They hypothesized that thiosulfate is the actual beneficial molecule. This is in line with findings from Marutani et al., who detected elevated cerebral thiosulfate, but not sulfide levels, in their murine model of brain I/R [47]. In contrast, Ravindran et al. reported elevated H_2_S levels in the brain after Na_2_S_2_O_3_ treatment and do not mention thiosulfate [45]. In murine LPS- and sepsis-induced acute lung injury, Sakaguchi et al. reported that Na_2_S_2_O_3_ treatment elevated both plasma and lung tissue thiosulfate and sulfide levels [50]. Shirozu et al. detected elevated plasma and liver thiosulfate levels after Na_2_S_2_O_3_ administration in their model, whereas sulfide levels were only elevated in the plasma for 6 h [49]. Thus, they also hypothesized that thiosulfate itself mediated the beneficial effects, in particular, that CSE^−/−^ mice in their study also were characterized by elevated thiosulfate levels and mirrored the beneficial effects of Na_2_S_2_O_3_ in wildtype animals [49]. Recent work from Marutani et al. also suggests that H_2_S catabolism, i.e., high SQR activity, which would naturally contribute to elevated thiosulfate levels, can contribute to cerebral hypoxia tolerance [67]. For further information on the potential therapeutic perspective for sulfide catabolism and the sulfide/thiosulfate interplay, the reader is referred to a recent review by Marutani and Ichinose [68]. Further potential downstream cellular signaling pathways for therapeutic effects of Na_2_S_2_O_3_, such as Nrf2 signaling, have been recently reviewed by Zhang et al. [69].

Na_2_S_2_O_3_ is an example for a sulfane sulfur-containing compound, which are regarded as a form of H_2_S storage, which can easily release this gasotransmitter in response to biological signals. Both reactive sulfur species (H_2_S and sulfane sulfur) always coexist in a biological system. Toohey has indicated that H_2_S is rather a biodegradation byproduct of sulfane sulfur-containing compounds. The author suggests that the sulfane sulfur compounds, which are present in cells at higher concentrations than H_2_S, are responsible for the observed biological effects attributed to H_2_S [70,71], which strengthens the speculation that thiosulfate is “more important” than H_2_S.

## 4. Clinical Perspective

Regardless of the underlying mechanism for the cytoprotective effects of Na_2_S_2_O_3_, an open question still exists as to whether there is a future for this clinically available compound in critical care. The pre-clinical results are promising, but have certain limitations: (1) only three out of the 19 studies presented in Table 1 investigated a mixture of male and female animals [13,46,52], in fact most of the studies were limited to male animals; (2) old age as a complication, which can affect both the expression of the endogenous H_2_S enzymes, as well as H_2_S levels [72], was not considered in any of the pre-clinical studies presented here; (3) of 12 in vivo studies presented in Table 1, only 4 studies included basic fluid resuscitation [47,48,49,56] and only 2 studies included resuscitation strategies that are common practice in critical care [13,53]. Thus, the translational relevance of the other studies remains to be proven.

Already in 1966, Paris et al. were able to achieve an increase in urine output by Na_2_S_2_O_3_ treatment in 16 out of 20 patients with severe burn-induced shock, which was even associated with alleviating the shock in these patients [73]. Currently (February 2022), 63 studies are listed on clinicaltrials.gov with the search term “sodium thiosulfate”. Most of these trials (34) look at Na_2_S_2_O_3_ as an adjuvant in chemotherapy, as an antidote for cisplatin intoxication (see Figure 3). Another 20 trials deal with diseases tied into calcium dysregulations (calciphylaxis, vascular calcification, calcinosis cutis), trying to make use of the calcium chelating properties of Na_2_S_2_O_3_ (see Figure 3). Three trials look at contact dermatitis and three further trials investigate endodontic disease each. One trial investigates the potential of Na_2_S_2_O_3_ for kidney transplantation. Only two of the listed trials are directly relevant to critical care: NCT03017963 “Safety and Tolerability of Sodium Thiosulfate in Patients With an Acute Coronary Syndrome (ACS) Undergoing Coronary Angiography Via Trans-radial Approach.”, and NCT02899364 “Sodium Thiosulfate to Preserve Cardiac Function in STEMI (GIPS-IV)”.

The results of the ACS trial have already been published [74]. A dose-escalation study with i.v. Na_2_S_2_O_3_ to a maximum dose of 15 g in two doses has been performed on 18 patients undergoing coronary angiography for ACS. A slight drop in systolic blood pressure was determined 1 h after the administration of the first dose, which could not be observed after the second dose. Two patients experienced brief periods of hypotension, one accompanied by mild nausea. No serious adverse events were observed [74]. The same group will investigate the potential benefit of an intervention with Na_2_S_2_O_3_ in a follow-up double-blind, randomized, placebo-controlled, multicenter trial (GIPS-IV), with a planned enrollment of 380 patients with ST-elevation myocardial infarction (STEMI) [75]. Patients will receive 12.5 g Na_2_S_2_O_3_ directly after admission and a second dose 6 h later. The primary endpoint is myocardial infarct size after 4 months; secondary endpoints include the effects of Na_2_S_2_O_3_ on peak CK-MB (creatine kinase muscle brain type, as an early measure of MI) during admission and left ventricular ejection fraction and NT-proBNP (N terminal pro brain natriuretic peptide, as a measure of heart insufficiency) levels at 4 months follow-up [75]. The group expects first results on the primary endpoint soon (“Q1 2022”), but no updates have been published yet.

Another interesting perspective for the clinical application of Na_2_S_2_O_3_ is the treatment of pre-eclampsia. Preliminary results of our own observational study DRKS0001771 (“Role of the oxytocin-receptor and the H_2_S-system in preeclampsia and HELLP-syndrome (NU-HOPE)”) confirm the dysregulation of endogenous H_2_S availability in pre-eclampsia, which has also been published previously [76,77]. Pre-eclampsia is not necessarily a critical illness, but can develop into one, in particular in case of “Hemolysis Elevated Liver Enzymes Low Platelet count (HELLP)” syndrome, thus it is relevant to be considered in the context of this mini review. The interaction between the endogenous H_2_S and oxytocin systems, which has recently been identified, might play a particularly important role in the molecular mechanism of pre-eclampsia and make the perspective of treatment with an H_2_S donor even more relevant. CSE^−/−^ and oxytocin receptor (OTR)^−/−^ mice show a reciprocal loss of other proteins in the heart [78,79]. Furthermore, resuscitated CSE^−/−^ mice had a particularly pronounced loss of cardiac OTR expression after bunt chest trauma, which could be restored by the administration of an exogenous H_2_S donor (GYY4137) [80]. Preliminary results from our observational trial confirm this interaction and dysregulation of the H_2_S- and OT-systems during pre-eclampsia (see Figure 4A). Furthermore, psychological trauma might play an important role in this context: a dysregulation and interaction of the H_2_S- and OT-systems has been shown in an animal model of early life stress [78], and placental CSE expression has been shown to be directly related to the patients’ childhood trauma load (see Figure 4B). Taken together, these results suggest that patients with underlying psychological trauma, in particular, might benefit from treatment with Na_2_S_2_O_3_.

COVID-19 patients might be another group that could profit from Na_2_S_2_O_3_ therapy. Na_2_S_2_O_3_ is a recognized drug devoid of major side effects, which attenuated murine acute lung injury [50] and cerebral ischemia/reperfusion injury [47]. It was also shown that Na_2_S_2_O_3_ significantly attenuated shock-induced impairment of lung mechanics and gas exchange in pigs after hemorrhagic shock [13]. Plasma H_2_S levels of survivors of COVID-19 pneumonia were significantly higher at day 1 and day 7 after admission in comparison to non-survivors [81]. These results suggest that H_2_S might be a valuable biomarker for the severity of COVID-19 infection on the one hand [81], and that exogenous administration of H_2_S might be a relevant therapeutic approach for these patients on the other hand [82]. Even though several groups have suggested Na_2_S_2_O_3_ as a therapeutic adjuvant in the therapy of COVID-19 patients [82,83], there currently are no registered clinical trials on the subject. In contrast, clinicaltrials.gov lists 13 clinical trials for the therapeutic potential of N-acetyl-cysteine (NAC) for COVID-19 patients. NAC is an antioxidant molecule, also able to elevate sulfide levels, which might have various benefits for SARS-CoV-2 [84].

## 5. Conclusions

Sodium thiosulfate (Na_2_S_2_O_3_) is a clinically approved H_2_S donor with a good safety profile. Pre-clinical experimental studies show that Na_2_S_2_O_3_ can mediate cyto-protection in various types of ischemia reperfusion injury, physical trauma, and inflammation. The first clinical trials investigating the effects of Na_2_S_2_O_3_ in myocardial infarct in humans are currently ongoing. Extending these clinical trials to other conditions would help uncover the full therapeutic potential for Na_2_S_2_O_3_ in critical care.

## Figures and Tables

**Figure 1 biomolecules-12-00543-f001:**
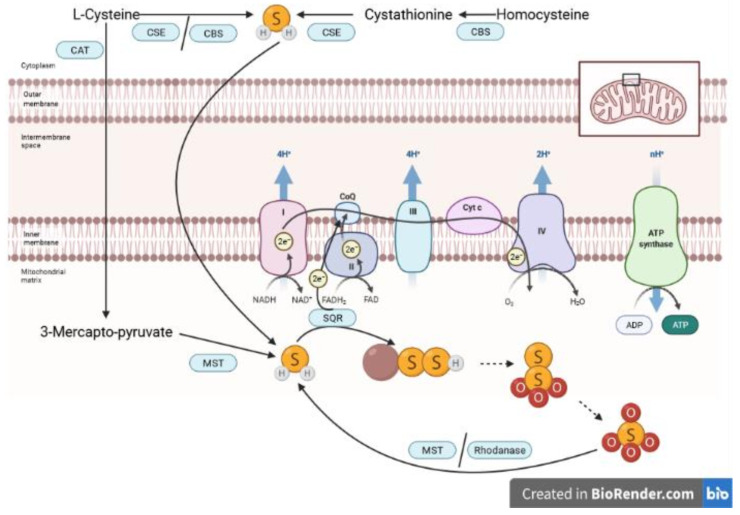
Sulfide generation and oxidation pathways. In the cytoplasm, cystathionine-γ-lyase (CSE) and cystathionine-β-synthase (CBS) facilitate H_2_S release from L-Cysteine and homocysteine. Cysteine-aminotransferase uses L-cysteine to form 3-mercapto-pyruvate, which is then used by 3-mercaptopyruvate-sulfurtransferase (MST) for mitochondrial H_2_S release. Sulfide quinone oxidoreductase (SQR) oxidizes H_2_S to persulfides in the mitochondria, and persulfides are further oxidized, which ultimately results in the formation of thiosulfate and sulfate. MST and rhodanese can re-generate H_2_S from thiosulfate, a process which can also happen non-enzymatically. Figure created in BioRender. Adapted from “Electron transport chain”, by BioRender.com (2022). Retrieved from https://app.biorender.com/biorender-templates (accessed on 15 March 2022).

**Figure 2 biomolecules-12-00543-f002:**
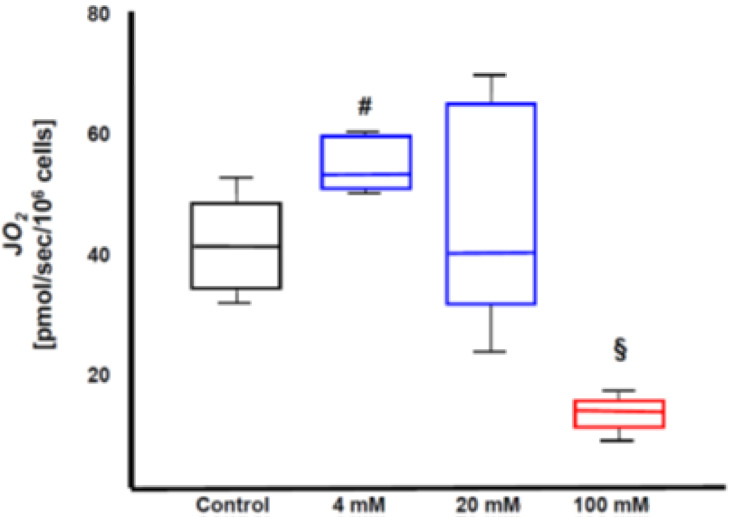
Effects of Na_2_S_2_O_3_ on mitochondrial respiration (oxidative phosphorylation as assessed by routine O_2_ consumption JO_2_) in cultured cortical neurons from fetal rat brains. Primary neuron cultures from the fronto-temporal cortex were prepared from fetal rat brains (embryonic day 18) and seeded on poly-L-lysine-coated culture flasks. The cells were grown in neurobasal medium, and complemented with B27 supplement, L-glutamine, and penicillin/streptomycin. At day 22–24 of culturing, cells were incubated with PBS (control) or 4, 20, or 100 mM Na_2_S_2_O_3_ for 4 h and harvested in respiration medium (MIR05: 0.5 mM EGTA, 3 mM MgCl2, 60 mM Lactobionic acid, 20 mM Taurine, 10 mM KH2PO4, 20 mM HEPES, 110 mM sucrose, 1 g/L bovine serum albumin). Mitochondrial oxygen consumption was determined after the addition of 10 mM pyruvate, 10 mM glutamate, 5 mM malate, 5 mM ADP, 10 µM cytochrome c, 10mM succinate, and uncoupling with 1.5 mM Carbonyl cyanide-4-(trifluoromethoxy) phenylhydrazone (FCCP). N = 6 measurements in each group. Data are median (IQR), # *p* < 0.01 and § *p* < 0.0001 vs. control. A low Na_2_S_2_O_3_ concentration (4 mM) increased, while a high concentration (100 mM) inhibited mitochondrial respiratory activity. Intermediate concentrations (20 mM) had variable effects.

**Figure 3 biomolecules-12-00543-f003:**
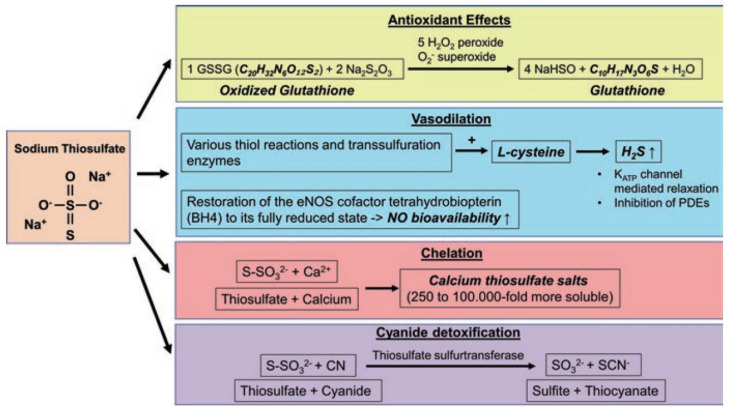
Biochemical effects of sodium thiosulfate. Na_2_S_2_O_3_ can reduce oxidized glutathione [34], mediate vaso-dilation [35], work as a calcium chelator [36], and work as an antidote for cyanide [37]. Figure taken from [38]. Copyright 2019 Springer Nature Switzerland AG. Reprinted with permission.

**Figure 4 biomolecules-12-00543-f004:**
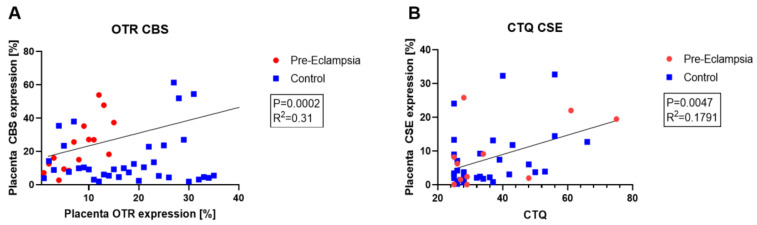
Preliminary results of the NU-HOPE observational trial. (**A**): Placental expression of the H_2_S-producting enzyme cystathionine-β-synthase (CBS) and oxytocin receptor (OTR) directly correlate. (**B**): The childhood trauma questionnaire score (CTQ) directly correlates with placental expression of the H_2_S-producing enzyme cystathionine-γ-lyase (CSE).

## Data Availability

Not applicable.

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
