# Peer review of "H2S in Critical Illness—A New Horizon for Sodium Thiosulfate?"

_biomolecules, 2022, doi:10.3390/biom12040543_

Round 1

Reviewer 1 Report

The Authors presented possible therapeutic potential of sodium thiosulfate (STS) in critical illness. In the introduction the authors present in detail the relationship of STS with hydrogen sulfide (H2S), supplementing the data with a detailed figure. However, there are some issues that need clarification. 

  • Please, specify in the table in the conditions column which disease model is meant, e.g. in addition to the information about passing LPS, the information that it is a sepsis model should be added
  • The text should explain what "Langendorff heart model" means.
  • H2S and its donors can act as vasodilators. This may increase hypotension during different types of shock. Please comment.
  • Authors stated that slow-releasing H2S donors also render them unlikely for clinical development, since they failed to exert organ protection. However, there is a number of reports showing positive results for GYY4137 in sepsis for example. 
  • Can STS act and release H2S at any pH, or only during hypoxia and pH decrease.
  • Please discuss the potential mechanism of action of STS for each of the indications presented in the work.
  • Are there data for STS administration in COVID-19. If not, in a work on STS activities, there should be no paragraph on COVID-19 (last paragraph)

Author Response

The Authors presented possible therapeutic potential of sodium thiosulfate (STS) in critical illness. In the introduction the authors present in detail the relationship of STS with hydrogen sulfide (H2S), supplementing the data with a detailed figure. However, there are some issues that need clarification.

We thank the reviewer for their suggestions to improve our paper.

    Please, specify in the table in the conditions column which disease model is meant, e.g. in addition to the information about passing LPS, the information that it is a sepsis model should be added

done, see modified table 1

    The text should explain what "Langendorff heart model" means.

done, l. 160-161

„Langendorff rat heart model (i.e. isolated heart with retrograde perfusion with a nutrient-rich oxygenated buffer)“

    H2S and its donors can act as vasodilators. This may increase hypotension during different types of shock. Please comment.

done, l. 123-126

„Even though the potential vaso-dilatory effects of H2S donors could aggravate shock-induced hypotension [7][7], other beneficial H2S effects might outweigh that risk. Still, the optimal timing and dosing window of H2S donors in these conditions must be evaluated carefully.“

    Authors stated that slow-releasing H2S donors also render them unlikely for clinical development, since they failed to exert organ protection. However, there is a number of reports showing positive results for GYY4137 in sepsis for example.

The beneficial effects of GYY4137 in sepsis models, that we are aware of, were only obtained in pre-clinical rodent models that did not include ICU measures. When GYY4137 was investigated in a translationally relevant porcine model of peritonitis with intensive care management, no clinically relevant benefit could be detected (Nußbaum et al., Shock. 2017, 48(2):175-184).

This has been clarified now in l. 135-139:

„However, translationally relevant in vivo results for slow-releasing H2S donors also render them unlikely for clinical development, since they failed to exert organ protection and/or even have adverse effects in these models so far [7,8]. in spite of many promising pre-clinical studies in non-resuscitated rodents.“

    Can STS act and release H2S at any pH, or only during hypoxia and pH decrease.

According to the literature, STS can release H2S only under hypoxic conditions, but not when oxygen is abundant (Olson et al. Am J Physiol Regul Integr Comp Physiol 305: R592–R603, 2013). H2S release is largely increased by reducing conditions (see edited l. 73 and 94). We are not aware of any studies investigating the pH-dependency of H2S release from STS.

    Please discuss the potential mechanism of action of STS for each of the indications presented in the work.

The putative mechanisms (e.g. anti-oxidant effects, anti-apoptotic effects, mitochondrial protection, anti-inflammatory effects) are discussed in the context of the results of the preclinical studies (e.g. see l. 165-166, l. 195, l. 223…) as well as mentioned in table 1.

    Are there data for STS administration in COVID-19. If not, in a work on STS activities, there should be no paragraph on COVID-19 (last paragraph)

Even though there are no direct data investigating STS administration in COVID-19, speculating on the potential effects could potentially lead to filling an important missing therapeutic gap with a clinically approved H2S donor with minimal side effects. As it is the other reviewers requested us to actually extend this section, thus we decided against deleting it.

Reviewer 2 Report

     Review of the paper entitled “H2S in critical illness – a new horizon for sodium thiosulfate?” by Tamara Merz, Oscar McCook, Cosima Brucker, Christiane Waller, Enrico Calzia1, Peter Radermacher, and Thomas Datzmann.

This review summarizes the current evidence for the therapeutic potential of sodium thiosulfate (Na2S2O3), a clinically approved H2S donor with minimal side effects, in many serious illness. The paper is interesting and well written.

My comment

I miss some interesting issues in this paper.

  • It is known that 3-mercaptosulfurtransferase (3-MST) produces H2S from 3- mercaptopyruvate, which is generated from L- or D-cysteine. The sulfur atom is transported from 3-mercaptopyruvate by MST in the form of sulfane sulfur (MST-SSH). It was shown that H2S could be released in the presence of dithiols [dihydrolipoic acid (DHLA), DTT, thioredoxin (Trx)]; however, in the presence of monothiols (GSH, cysteine) no release of H2S was observed [Mikami, Y., Shibuya, N., Kimura, Y., Nagahara, N., Ogasawara, Y. and Kimura, H. (2011) Thioredoxin and DHLA acid are required for 3-mercaptopyruvate sulfurtransferase to produce hydrogen sulfide. Biochem. J. 439, 479–485].
  • Mitochondrial H2S oxidation is catalysed by the following enzymes: sulfide quinone oxidoreductase (SQR) persulfide dioxygenase (ETHE1) and rhodanese (thiosulfate sulfur ransferase; TST). A physiological acceptor of sulfane sulfur from SQR-SSH has not been identified unequivocally, yet. Some authors have postulated that human SQR utilizes sulfite as persulfide acceptor yielding thiosulfate as a product. Other authors have demonstrated that in addition to sulfite, GSH functions as a persulfide acceptor for human SQR leading to GSSH [Libiad M, Yadav PK, Vitvitsky V, Martinov M, Banerjee R. Organization of the human mitochondrial hydrogen sulfide oxidation pathway. J Biol Chem. 2014 Nov 7;289(45):30901-10. doi: 10.1074/jbc.M114.602664. Epub 2014 Sep 15. PMID: 25225291; PMCID: PMC4223297]. It is also not excluded that there are SQR persulfide acceptors other than sulfite and GSH (e.g. DHLA, Trx, cysteine), which after accepting sulfane sulfur can be reduced by GSH [Kabil O, Banerjee R. Enzymology of H2S biogenesis, decay and signaling. Antioxid Redox Signal. 2014 Feb 10;20(5):770-82. doi: 10.1089/ars.2013.5339. Epub 2013 Jun 7. PMID: 23600844; PMCID: PMC3910450].
  • The relationship between H2S and sulfane sulfur compounds. Compounds containing sulfane sulfur are also regarded as a form of H2S storage, which can easily release this gasotransmitter in response to biological signals. Both reactive sulfur species (H2S and sulfane sulfur) always coexist in biological system. Toohey has indicated that H2S is rather a biodegradation byproduct of sulfane-sulfur-containing compounds. His review paper analyzes the possible misinterpretations in the literature and compares the relative properties of H2S and sulfane sulfur as signaling agents. The author suggests that it is the sulfane sulfur compounds rather than the H2S are responsible for the observed biological effects. It should be remembered that sulfane sulfur compounds are present in cells at higher concentrations than H2S [Toohey JI. Sulphane sulphur in biological systems: a possible regulatory role. Biochem J. 1989 Dec 15;264(3):625-32. doi: 10.1042/bj2640625. Erratum in: Biochem J 1990 May 1;267(3):843. PMID: 2695062; PMCID: PMC1133633]. Na2S2O3 is a sulfane sulfur-containing compound. It therefore appears that thiosulfate is “more important” than hydrogen sulfide. Is it really so?
  • Some authors note that hemoproteins are important players in the reactions of reactive sulfur species significantly contributing to Na2S2O3 production from H2S (for a review: Bilska-Wilkosz A, Iciek M, Górny M, Kowalczyk-Pachel D. The Role of Hemoproteins: Hemoglobin, Myoglobin and Neuroglobin in Endogenous Thiosulfate Production Processes. Int J Mol Sci. 2017 Jun 20;18(6):1315. doi: 10.3390/ijms18061315. PMID: 28632164; PMCID: PMC5486136].

I am asking the Authors to include the topics in their publication I raised above.

Other comments

     The Authors wrote:

“COVID-19 patients might be another group profiting from Na2S2O3 therapy: Plasma H2S levels of survivors of COVID-19 pneumonia were significantly higher at day 1 and day 7 after admission in comparison to non-survivors [70]. These results suggest that H2S might be a valuable biomarker for the severity of COVID-19 infection on the one hand [70], and that exogenous administration of H2S might be a relevant therapeutic approach for these patients on the other hand [71]”.

The Authors cited two articles here, namely:

Renieris, G.; Katrini, K.; Damoulari, C.; Akinosoglou, K.; Psarrakis, C.; Kyriakopoulou, M.; Dimopoulos, G.; Lada, M.; 529 Koufargyris, P.; Giamarellos-Bourboulis, E.J. Serum Hydrogen Sulfide and Outcome Association in Pneumonia by the SARS-CoV-2 Coronavirus. Shock (Augusta, Ga.) 2020, 54, 633–637, doi:10.1097/SHK.0000000000001562 and Datzmann T, Merz T, McCook O, Szabo C, Radermacher P. H2S as a Therapeutic Adjuvant Against COVID-19: Why and How? Shock. 2021 Nov 1;56(5):865-867. doi: 10.1097/SHK.0000000000001723. PMID: 33443365; PMCID: PMC8518209.

     However, I want to note that these publications are not about Na2S2O3 at all, only about H2S. In my opinion, these are not adequately cited papers. It would be better if the Authors wrote that Na2S2O3 is a potential candidate as a drug supporting the treatment of COVID-19 patients, because it is known that Na2S2O3 is a recognized drug devoid of major side effects, which attenuated murine acute lung injury and cerebral ischemia/reperfusion injury. It was also shown that Na2S2O3 significantly attenuates shock-induced impairment of lung mechanics and gas exchange in animals. [Datzmann T, Hoffmann A, McCook O, Merz T, Wachter U, Preuss J, Vettorazzi S, Calzia E, Gröger M, Kohn F, Schmid A, Denoix N, Radermacher P, Wepler M. Effects of sodium thiosulfate (Na2S2O3) during resuscitation from hemorrhagic shock in swine with preexisting atherosclerosis. Pharmacol Res. 2020 Jan;151:104536. doi: 10.1016/j.phrs.2019.104536. Epub 2019 Nov 14. PMID: 31734346]. It should be noted, however, that so far Na2S2O3 has not been studied in SARS-CoV-2 patients.

Section title 2 is: Biology of H2S and S2O3

It should be: Biology of H2S and S2O32- or Biology of H2S and Na2S2O3 (in my opinion).

Author Response

This review summarizes the current evidence for the therapeutic potential of sodium thiosulfate (Na2S2O3), a clinically approved H2S donor with minimal side effects, in many serious illness. The paper is interesting and well written.

We thank the reviewer for their interest in our work and their suggestions to improve our paper.

My comment

I miss some interesting issues in this paper.

    It is known that 3-mercaptosulfurtransferase (3-MST) produces H2S from 3- mercaptopyruvate, which is generated from L- or D-cysteine. The sulfur atom is transported from 3-mercaptopyruvate by MST in the form of sulfane sulfur (MST-SSH). It was shown that H2S could be released in the presence of dithiols [dihydrolipoic acid (DHLA), DTT, thioredoxin (Trx)]; however, in the presence of monothiols (GSH, cysteine) no release of H2S was observed [Mikami, Y., Shibuya, N., Kimura, Y., Nagahara, N., Ogasawara, Y. and Kimura, H. (2011) Thioredoxin and DHLA acid are required for 3-mercaptopyruvate sulfurtransferase to produce hydrogen sulfide. Biochem. J. 439, 479–485].

    Mitochondrial H2S oxidation is catalysed by the following enzymes: sulfide quinone oxidoreductase (SQR) persulfide dioxygenase (ETHE1) and rhodanese (thiosulfate sulfur ransferase; TST). A physiological acceptor of sulfane sulfur from SQR-SSH has not been identified unequivocally, yet. Some authors have postulated that human SQR utilizes sulfite as persulfide acceptor yielding thiosulfate as a product. Other authors have demonstrated that in addition to sulfite, GSH functions as a persulfide acceptor for human SQR leading to GSSH [Libiad M, Yadav PK, Vitvitsky V, Martinov M, Banerjee R. Organization of the human mitochondrial hydrogen sulfide oxidation pathway. J Biol Chem. 2014 Nov 7;289(45):30901-10. doi: 10.1074/jbc.M114.602664. Epub 2014 Sep 15. PMID: 25225291; PMCID: PMC4223297]. It is also not excluded that there are SQR persulfide acceptors other than sulfite and GSH (e.g. DHLA, Trx, cysteine), which after accepting sulfane sulfur can be reduced by GSH [Kabil O, Banerjee R. Enzymology of H2S biogenesis, decay and signaling. Antioxid Redox Signal. 2014 Feb 10;20(5):770-82. doi: 10.1089/ars.2013.5339. Epub 2013 Jun 7. PMID: 23600844; PMCID: PMC3910450].

    The relationship between H2S and sulfane sulfur compounds. Compounds containing sulfane sulfur are also regarded as a form of H2S storage, which can easily release this gasotransmitter in response to biological signals. Both reactive sulfur species (H2S and sulfane sulfur) always coexist in biological system. Toohey has indicated that H2S is rather a biodegradation byproduct of sulfane-sulfur-containing compounds. His review paper analyzes the possible misinterpretations in the literature and compares the relative properties of H2S and sulfane sulfur as signaling agents. The author suggests that it is the sulfane sulfur compounds rather than the H2S are responsible for the observed biological effects. It should be remembered that sulfane sulfur compounds are present in cells at higher concentrations than H2S [Toohey JI. Sulphane sulphur in biological systems: a possible regulatory role. Biochem J. 1989 Dec 15;264(3):625-32. doi: 10.1042/bj2640625. Erratum in: Biochem J 1990 May 1;267(3):843. PMID: 2695062; PMCID: PMC1133633]. Na2S2O3 is a sulfane sulfur-containing compound. It therefore appears that thiosulfate is “more important” than hydrogen sulfide. Is it really so?

    Some authors note that hemoproteins are important players in the reactions of reactive sulfur species significantly contributing to Na2S2O3 production from H2S (for a review: Bilska-Wilkosz A, Iciek M, Górny M, Kowalczyk-Pachel D. The Role of Hemoproteins: Hemoglobin, Myoglobin and Neuroglobin in Endogenous Thiosulfate Production Processes. Int J Mol Sci. 2017 Jun 20;18(6):1315. doi: 10.3390/ijms18061315. PMID: 28632164; PMCID: PMC5486136].

I am asking the Authors to include the topics in their publication I raised above.

These are indeed very interesting issues, which have now been added to our paper accordingly.

l.53-58: „3-MST produces H2S from 3- mercaptopyruvate, which is generated from cysteine (see figure 1). The sulfur atom is transported from 3-mercaptopyruvate by MST in the form of sulfane sulfur (MST-SSH). It was shown that H2S could be released in the presence of dithiols [dihydrolipoic acid (DHLA), DTT, thioredoxin (Trx)]; however, in the pres-ence of monothiols (GSH, cysteine) no release of H2S was observed [17].“

l. 63-69: „A physiological acceptor of sulfane sulfur from SQR-SSH has not been identified une-quivocally, yet. Some authors have postulated that human SQR utilizes sulfite as per-sulfide acceptor yielding thiosulfate as a product. Other authors have demonstrated that in addition to sulfite, GSH functions as a persulfide acceptor for human SQR leading to GSSH [21]. It is also not excluded that there are SQR persulfide accep-tors other than sulfite and GSH (e.g. DHLA, Trx, cysteine), which after accepting sul-fane sulfur can be reduced by GSH [22]“

l. 71-72: „Hemoproteins might also contribute to thiosulfate production from H2S [23]“

l. 73-76: „H2S can be regenerated from exogenously administered thiosulfate by rhondanease and, reportedly, 3-MST [14](see Figure 1), which is not surprising when consider-ing that both thiosulfate and 3-mercaptopyruvate are, chemically speaking, sulfane sulfurs .

l. 291-298: „Na2S2O3 is an example for a sulfane sulfur-containing compound, which are re-garded as a form of H2S storage, which can easily release this gasotransmitter in re-sponse to biological signals. Both reactive sulfur species (H2S and sulfane sulfur) al-ways coexist in a biological system. Toohey has indicated that H2S is rather a biodeg-radation byproduct of sulfane-sulfur-containing compounds. The author suggests that the sulfane sulfur compounds, which are present in cells at higher concentrations than H2S, are responsible for the observed biological effects attributed to H2S. [70,71], which strengthens the speculation that thiosulfate is “more important” than H2S.

Other comments

     The Authors wrote:

“COVID-19 patients might be another group profiting from Na2S2O3 therapy: Plasma H2S levels of survivors of COVID-19 pneumonia were significantly higher at day 1 and day 7 after admission in comparison to non-survivors [70]. These results suggest that H2S might be a valuable biomarker for the severity of COVID-19 infection on the one hand [70], and that exogenous administration of H2S might be a relevant therapeutic approach for these patients on the other hand [71]”.

The Authors cited two articles here, namely:

Renieris, G.; Katrini, K.; Damoulari, C.; Akinosoglou, K.; Psarrakis, C.; Kyriakopoulou, M.; Dimopoulos, G.; Lada, M.; 529 Koufargyris, P.; Giamarellos-Bourboulis, E.J. Serum Hydrogen Sulfide and Outcome Association in Pneumonia by the SARS-CoV-2 Coronavirus. Shock (Augusta, Ga.) 2020, 54, 633–637, doi:10.1097/SHK.0000000000001562 and Datzmann T, Merz T, McCook O, Szabo C, Radermacher P. H2S as a Therapeutic Adjuvant Against COVID-19: Why and How? Shock. 2021 Nov 1;56(5):865-867. doi: 10.1097/SHK.0000000000001723. PMID: 33443365; PMCID: PMC8518209.

     However, I want to note that these publications are not about Na2S2O3 at all, only about H2S. In my opinion, these are not adequately cited papers. It would be better if the Authors wrote that Na2S2O3 is a potential candidate as a drug supporting the treatment of COVID-19 patients, because it is known that Na2S2O3 is a recognized drug devoid of major side effects, which attenuated murine acute lung injury and cerebral ischemia/reperfusion injury. It was also shown that Na2S2O3 significantly attenuates shock-induced impairment of lung mechanics and gas exchange in animals. [Datzmann T, Hoffmann A, McCook O, Merz T, Wachter U, Preuss J, Vettorazzi S, Calzia E, Gröger M, Kohn F, Schmid A, Denoix N, Radermacher P, Wepler M. Effects of sodium thiosulfate (Na2S2O3) during resuscitation from hemorrhagic shock in swine with preexisting atherosclerosis. Pharmacol Res. 2020 Jan;151:104536. doi: 10.1016/j.phrs.2019.104536. Epub 2019 Nov 14. PMID: 31734346]. It should be noted, however, that so far Na2S2O3 has not been studied in SARS-CoV-2 patients.

done (l. 367-378): „Na2S2O3 is a recognized drug devoid of major side effects, which attenuated murine acute lung injury [50]and cerebral ischemia/reperfusion injury [47]. It was also shown that Na2S2O3 significantly attenuated shock-induced impairment of lung me-chanics and gas exchange in pigs after hemorrhagic shock [13]. [...] Even though several groups have suggested Na2S2O3 as a therapeutic adjuvant in the therapy of Covid-19 patients [82,83], there currently are no registered clinical trials on the subject. “

Section title 2 is: Biology of H2S and S2O3

It should be: Biology of H2S and S2O32- or Biology of H2S and Na2S2O3 (in my opinion).

done

Reviewer 3 Report

The aim of the work is to show the promising role of Na2S2O3 in the acute management of critical illness and its therapeutic potential. 

The manuscript is hard to read, its long, rather detailed descriptions should be divided into subsections with clear conclusions (see chapter 3). It should be redrafted.

The manuscript requires some changes / additions:

l.49. The enzymes mentioned were known much earlier.

l.50-51 This statement is not clear - please specify.

l.53 “small amounts of H2S” - please specify the concentration range

l.57 not “rhondanase” but rhodanese or more precisely thiosulfate sulfurtransferase.

l.69-70 “H2S accumulates in the cell” - not quite rightly said because this molecule diffuses through membranes

l.71 “the degeneration of preexisting thiosulfate” – why degeneration?

In the description under Figure 2, the concentration unit is sometimes given correctly (mM) and sometimes not (mMol)

The effects presented in Figure 3 should be supported by citing the relevant literature.

l.182-183 – it is unconfirmed and should not be suggested

l.200 - (. These studies - unnecessary characters

l. 204 – “his benefit of Na2S2O3 was lost [54] (under review)” - should be deleted as unpublished

Author Response

The aim of the work is to show the promising role of Na2S2O3 in the acute management of critical illness and its therapeutic potential.

We thank the reviewer for their suggestions to improve our paper.

The manuscript is hard to read, its long, rather detailed descriptions should be divided into subsections with clear conclusions (see chapter 3). It should be redrafted.

According to the reviewer’s suggestion, chapter 3 has been divided into subsections.

The manuscript requires some changes / additions:

l.49. The enzymes mentioned were known much earlier.

The statement has been misunderstood and thus has been edited (now l. 52): „since then three different enzymes have been identified to be able to release H2S endogenously: cystathionine-γ-lyase (CSE), cystathi-onine-β-synthase (CBS) and 3-MST.“

l.50-51 This statement is not clear - please specify.

done: „H2S oxidation represents another level of the regulation of endogenous H2S availabil-ity, besides its endogenous production.“ (now l. 58-60)

l.53 “small amounts of H2S” - please specify the concentration range

A concentration range (nanomolar to low micromolar) has now been added to the statement (l. 62). It has to be considered, that there is a very high variability (i.e. within three orders of magnitude) in reported H2S levels, given the difficulty of measuring this highly reactive and volatile molecule (McCook et al., Nitric Oxide. 2014;41:48-61.). Thus reported concentrations are not reliable and can only be interpreted properly within the limitations of the used methodology.

l.57 not “rhondanase” but rhodanese or more precisely thiosulfate sulfurtransferase.

typo has been corrected

l.69-70 “H2S accumulates in the cell” - not quite rightly said because this molecule diffuses through membranes

has been edited: „cellular H2S concentrations rise“ (now l. 88)

l.71 “the degeneration of preexisting thiosulfate” – why degeneration?

Statement was unclear and thus has been deleted

In the description under Figure 2, the concentration unit is sometimes given correctly (mM) and sometimes not (mMol)

has been corrected

The effects presented in Figure 3 should be supported by citing the relevant literature.

done

l.182-183 – it is unconfirmed and should not be suggested

The statement clearly says, that this is a speculation. We decided against deleting the statement, to hopefully inspire more research in that direction.

l.200 - (. These studies - unnecessary characters

has been corrected

l. 204 – “his benefit of Na2S2O3 was lost [54] (under review)” - should be deleted as unpublished

The journal specifically allows for citation of unpublished work. As the paper is already submitted to a journal, we would like to leave the statement as is. For the reviewer’s convenience, we are attaching current versions of both papers under review that were cited in this work (Messerer 2022 and Gröger 2022) to this response letter. The work from Messerer et al. has been suggested for acception after minor revisions, the work from Gröger et al. has been endorsed for publication by two reviewers after rebuttal, with one reviewer’s response still pending.

Reviewer 4 Report

This is a rather solid and comprehensive review of several authors specialized in "H2S" field. The authors briefly described the molecular details of H2S production by transsulfuration process and metabolism and concentrated on the role of sodium thiosulfate (STS) in various illnesses and pathological conditions including inflammation. The choice is perfect because this compound is quite safe and cheap and FDA approved substance. However, I want to make a few minor remarks.

  1. It will be nice to cite at least one basic paper of Dr.Kimura;
  2. It is necessary to mention that there are fast (e.g. STS) and slow (GYY4137) donors of H2S and they may have different effects.
  3. In the last years STS and other donors of H2S (e.g.acetyl-cystein) attracted a lot of attention and were used as indicators of disease severity and treatment of COVID-19 patients. (see pioneer works of Citi et al., 2020; Evgen'ev and Frenkel, 2020). In the present review there are only a couple of sentences describing the role of H2S dinors in the disease. I think it is necessary to extend this section using multiple experimental and clinical data on the protective role of STS in COVID-19 patients.

However

Author Response

This is a rather solid and comprehensive review of several authors specialized in "H2S" field. The authors briefly described the molecular details of H2S production by transsulfuration process and metabolism and concentrated on the role of sodium thiosulfate (STS) in various illnesses and pathological conditions including inflammation. The choice is perfect because this compound is quite safe and cheap and FDA approved substance. However, I want to make a few minor remarks.

We thank the reviewer for their appreciation of our work and suggestions to improve our paper.

    It will be nice to cite at least one basic paper of Dr.Kimura;

done (l. 51 and l. 58)

    It is necessary to mention that there are fast (e.g. STS) and slow (GYY4137) donors of H2S and they may have different effects.

We are not aware that STS has been characterized as neither fast nor slow releasing H2S donor. Nonetheless, we inserted a statement on the potentially contrasting effects of fast and slow releasing donors (l.130-135): „H2S-releasing salts cause rapid high, potentially toxic, peak H2S concentrations, which dissipate quickly [28], can have pro-inflammatory effects [29] and damage the mitochondria [20,30]. […] In contrast, slow H2S-releasing donors seem to have different effects, in that they rather seem to ameliorate inflammation [29].

    In the last years STS and other donors of H2S (e.g.acetyl-cystein) attracted a lot of attention and were used as indicators of disease severity and treatment of COVID-19 patients. (see pioneer works of Citi et al., 2020; Evgen'ev and Frenkel, 2020). In the present review there are only a couple of sentences describing the role of H2S dinors in the disease. I think it is necessary to extend this section using multiple experimental and clinical data on the protective role of STS in COVID-19 patients.

The section has been extended according to the reviewer’s suggestions (l. 376-381): „Even though several groups have suggested Na2S2O3 as a therapeutic adjuvant in the therapy of Covid-19 patients [82,83], there currently are no registered clinical trials on the subject. In contrast, clinicaltrials.gov lists 13 clinical trials for the thera-peutic potential of N-acetyl-cysteine (NAC) for Covid-19 patients. NAC is an antioxi-dant molecule, also able to elevate sulfide levels, which might have various benefits for SARS-CoV-2 [84].

However

Unfortunately, it seems like some of the reviewer’s comments might be missing here, thus we cannot respond to whatever got lost.